# Non-Coding RNAs as Sensors of Oxidative Stress in Neurodegenerative Diseases

**DOI:** 10.3390/antiox9111095

**Published:** 2020-11-08

**Authors:** Ana Gámez-Valero, Anna Guisado-Corcoll, Marina Herrero-Lorenzo, Maria Solaguren-Beascoa, Eulàlia Martí

**Affiliations:** 1Department de Biomedicina, Facultat de Medicina i Ciències de la Salut, Institut de Neurociències, Universitat de Barcelona, C/Casanova 143, 08036 Barcelona, Spain; a.gamez@ub.edu (A.G.-V.); annaguisado@ub.edu (A.G.-C.); marina.herrero@ub.edu (M.H.-L.); m.solaguren-beascoa@ub.edu (M.S.-B.); 2Centro de Investigación Biomédica en Red de Epidemiología y Salud Pública (CIBERESP), Ministerio de Ciencia Innovación y Universidades, 28046 Madrid, Spain

**Keywords:** oxidative stress, neurodegeneration, ncRNA, miRNA, tRNA fragments, lncRNA, circRNA

## Abstract

Oxidative stress (OS) results from an imbalance between the production of reactive oxygen species and the cellular antioxidant capacity. OS plays a central role in neurodegenerative diseases, where the progressive accumulation of reactive oxygen species induces mitochondrial dysfunction, protein aggregation and inflammation. Regulatory non-protein-coding RNAs (ncRNAs) are essential transcriptional and post-transcriptional gene expression controllers, showing a highly regulated expression in space (cell types), time (developmental and ageing processes) and response to specific stimuli. These dynamic changes shape signaling pathways that are critical for the developmental processes of the nervous system and brain cell homeostasis. Diverse classes of ncRNAs have been involved in the cell response to OS and have been targeted in therapeutic designs. The perturbed expression of ncRNAs has been shown in human neurodegenerative diseases, with these changes contributing to pathogenic mechanisms, including OS and associated toxicity. In the present review, we summarize existing literature linking OS, neurodegeneration and ncRNA function. We provide evidences for the central role of OS in age-related neurodegenerative conditions, recapitulating the main types of regulatory ncRNAs with roles in the normal function of the nervous system and summarizing up-to-date information on ncRNA deregulation with a direct impact on OS associated with major neurodegenerative conditions.

## 1. Introduction

Neurodegenerative disorders are a heterogeneous class of diseases characterized by a slow, progressive neuronal dysfunction and loss. The etiology of this class of diseases has not been fully elucidated, with highly complex and multifactorial causes participating in disease onset and progression. Nevertheless, several lines of evidence suggest that oxidative stress (OS) has a central role in this type of disorders. OS is the result of an imbalance between the increase in reactive oxygen species (ROS) and the production of antioxidants to eliminate them. A variety of mechanisms involving oxidative damage have been described in neurodegenerative conditions, including mitochondrial dysfunction, the oxidation of nucleic acids, proteins and lipids, activation of glial cells, apoptosis, cytokines production and inflammatory responses. The identification of molecular modifiers of the endogenous antioxidant activity is key to boost novel therapeutic designs [1].

The advent of high-throughput sequencing technologies revealed that the human genome is pervasively transcribed; with a large proportion corresponding to non-coding RNA (ncRNA) transcripts, the most abundant class of RNAs in cells [2,3]. ncRNAs have been classified according to their length and activity [4] with many classes revealed as essential regulators of gene expression through a variety of mechanisms [5].

ncRNAs are especially abundant and diverse in the central nervous system (CNS), showing a highly specific and dynamic temporal and spatial expression pattern that guides neurodevelopmental processes, including neural stem cell proliferation and differentiation [6,7]. In addition to their role in developmental processes, ncRNAs are key for the maintenance and normal functioning of adult, post-mitotic neurons. Most studies have been focused on microRNAs (miRNAs), the best-known class of ncRNAs. Although the roles for a progressively increasing number of ncRNAs are continuously being uncovered, the biological functions of the vast majority remain unknown.

While the precise expression levels of ncRNAs are key for the normal function of the CNS, the deregulation of ncRNA pathways have been linked to human disease, including neurodegenerative conditions [8,9,10,11]. In the present manuscript, we will provide an overview on the functional association between ncRNAs and the response to OS, a major hallmark in neurodegenerative disorders, and we will review the OS–ncRNA axis in major neurodegenerative conditions.

## 2. Oxidative Stress in Neurodegenerative Diseases

### 2.1. Reactive Oxygen Species Production and Activities

Reactive oxygen species (ROS), including hydrogen peroxide (H_2_O_2_) and free radicals such as superoxide anion (O_2_^−^) and hydroxyl radical (OH^−^), are the consequence of aerobic metabolism and endogenously produced as by-products of diverse enzymatic reactions, including those of the mitochondrial respiratory chain. During mitochondrial electron transport, O_2_ reduction results in superoxide (O_2_^•−^) that is detoxified to H_2_O_2_ by the mitochondrial manganese superoxide dismutase, and H_2_O_2_ can be converted to the hydroxyl radical (OH^−^). Another important source of ROS is the β-oxidation occurring in peroxisomes, whose main product is H_2_O_2_ [12,13]. Endogenously, ROS are generated following immune cell activation and ischemia or in ageing.

ROS can be also produced by non-enzymatic reactions, as those initiated by ionizing radiation, or as a result of the interaction between redox-active metals and oxygen species. Free iron (Fe^2+^) reacts with H_2_O_2_ (Fenton reaction) generating very reactive and damaging hydroxyl radicals. Other chemical reactions involving superoxide lead to Fe^2+^ production, which in turn affects redox recycling [14]. Calcium, which is an important signaling molecule, is indirectly involved in the generation of ROS by enhancing the electric flow into the respiratory chain and by stimulating the nitric oxide synthetase that produces nitric oxide. In addition, exogenous sources of ROS exist, including air pollution, cigarette smoke or heavy metals. These exogenous products are incorporated by cells and metabolized into free radicals [15,16].

At low concentrations, ROS can serve as relevant second messengers in cell signaling and are vital to human health. Examples of activities regulated by ROS at low to moderate concentrations include the support of cell proliferation and survival pathways in host defense cells [17] and the regulation of ATP production through the activation of uncoupling proteins [18]. Nevertheless, the increased and sustained production of ROS induces a pathological chain reaction that results in damage to DNA, lipids and proteins, overall producing progressive cell dysfunction [19,20]. To maintain physiological redox balance, cells have endogenous antioxidant defenses regulated at the transcriptional level by Nrf2/ARE [21]. The imbalance between the formation of ROS and the antioxidant capacity of cells to detoxify ROS or repair the resulting damage may eventually lead to many chronic diseases, such as atherosclerosis, cancer, diabetics, and neurodegenerative diseases.

### 2.2. Oxidative Stress in Neurodegeneration

Neurodegenerative diseases are characterized by progressive and chronic cell dysfunction and death that frequently starts affecting specific regions of the central nervous system (CNS), worsening over time and impacting more regions in a predictable fashion. Neuronal loss is often associated with protein misfolding and aggregation both in the extracellular space and intracellularly, in specific cells (Figure 1). While many neurons cope with sustained ROS increases, there are particular cell populations that show varying degrees of susceptibility to OS, a phenomenon that has been called differential vulnerability [22].

Although differential vulnerability is influenced by the etiology of each neurodegenerative condition; shared unique features generally render neuronal cells particularly susceptible. Compared to other cells, neurons are more dependent on mitochondrial oxidative phosphorylation, to fulfill their high-energy demands. Neurons are exposed to high O_2_ concentrations and consume 20% of the body oxygen [23]. About 1–2% of the consumed O_2_ is converted into ROS, with this percentage dramatically increasing with age [24]. In addition, neurons are enriched in metal ions, which are accumulated along aging and catalyze ROS formation; and are also rich in fatty acids that are prone to oxidation. This is accompanied by a decreased concentration of antioxidant enzymes in neurons compared with other cells and tissues, resulting in impaired antioxidant capacities in the brain [25].

OS is considered an important age-dependent factor. During the process of ageing, the capacity of neurons to counteract ROS accumulation diminishes, rendering them more susceptible to neurodegenerative conditions such as Alzheimer’s disease (AD) and Parkinson’s disease (PD) [26]. The excess of ROS produces the oxidation and misfunction of biomolecules [21,27,28]. The ROS mediated oxidation of amino-acid residues in proteins results in the introduction of carbonyl groups, which is a good estimate of OS extent [29,30,31]. ROS also produce lipid peroxidation and peroxy-radicals initiate a chain reaction leading to the formation of breakdown products such as 4-hydroxy-2, 3-nonenal (HNE) [32]. HNE can modify proteins and produce detrimental effects, including the dysregulation of intracellular calcium signaling which ultimately induces an apoptotic cascade in neurons [33,34,35]. OS damage in DNA involves hydroxylation, carbonylation and nitration [36,37] and is characterized by increased levels of 8-hydroxy-2-deoxyguanosine and 8-hydroxyguanosine [38,39]. OS can produce DNA strand breaks, which agrees with increased carbonyls in the nuclei, in neurodegenerative diseases. A non-enzymatic reaction of sugars with protein deposits produces advanced glycation end-products that are neurotoxic and pro-inflammatory molecules [40,41]. The accumulation of these products in neurodegenerative diseases is caused by an accelerated oxidation of glycated proteins [42].

The aggregation of misfolded proteins forming high-ordered insoluble fibrils is common to diverse neurodegenerative diseases and participates in neurotoxicity [43], although the underlying mechanism is still uncertain [44]. Abnormal folding is thought to be influenced both by genetic and environmental factors, with compelling evidence supporting a tight effect of OS [45]. Proteins modified by accumulated ROS form aggregates, which inhibit the proteasome, the major machinery for the removal of misfolded and aberrant proteins [46]. The normal function of the proteasome is essential for the timely removal of oxidized proteins and the maintenance of normal cell homeostasis, but the disruption of this process leads to protein accumulation and cell death [47,48]. 

Finally, the generation of ROS is regulated by redox-sensitive metals and therefore the disruption of metal homeostasis in the brain may also lead to the uncontrolled formation of ROS. Specifically, increased exposure to metals like iron, copper and zinc has been associated to neurodegenerative conditions through the disruption of the mechanisms that regulate their activity.

## 3. Regulatory Non-Coding RNAs in the Normal Function of the Nervous System

The human genome is almost transcribed in its entirety, from both strands-sense and reverse-, in a very tissue-, cell-, organ-environmental-specific way. Nevertheless, as descried in the Human Genome Project, only the 1–5% of it is transcribed into protein-coding genes [49]. The phase III (2012–2017) of the ENCODE (Encyclopedia of DNA Elements) project identified around 20,000 protein-coding genes and almost 40,000 are non-coding genes transcribed into ncRNA [50]. The increasing number of functional ncRNA in complex organisms has been proposed as the answer to the inconsistent correlation between the complexity of an organism and the number of protein-coding genes [51].

NcRNAs are classified as *housekeeping* RNAs and *regulatory* ncRNAs. Housekeeping ncRNA are highly expressed constitutively and ubiquitously and play crucial roles in routine cell maintenance; they include transfer RNAs (tRNAs), ribosomal RNAs (rRNAs), small nuclear RNAs (snRNAs) and small nucleolar RNAs (snoRNAs). Regulatory ncRNAs, which are the focus of the present review, are characterized by a highly regulated and specific expression pattern, by their activity as transcriptional and post-transcriptional gene expression modulators and their high sensitivity to cell needs [52]. Regulatory ncRNAs can be classified based on their length as long non-coding RNAs (lncRNAs; >200 nucleotides) and short/small non-coding RNAs (sncRNAs; <200 nucleotides, usually between 18 and 40 nt) [11]. Discoveries over the last 40 years have elucidated diverse functions for ncRNAs involving the regulation of mRNA expression stability and translation, including splicing regulation, chromosome maintenance and segregation, and even the transmission of regulatory signals.

In particular, RNA biology is of paramount significance in the CNS. The development of CNS and brain morphogenesis is a very complex spatiotemporal-organized process and requires an exquisite regulation of stem cell proliferation and differentiation. The brain is the organ showing the highest diversity of regulatory ncRNAs [7,53,54], and their dynamic expression and activity underlies CNS complex functions including neurodevelopment, synapse function, cell polarity and maturation, and the responses to intra- and extracellular stimuli [6]. In this section, we outline the role of the main classes of regulatory ncRNAs in the development and correct function of the nervous system. We have focused on main regulatory ncRNAs with (1) reported roles in OS management and (2) perturbed activity in neurodegenerative conditions.

### 3.1. microRNAs

microRNAs (miRNAs, ~21 nt), the most studied class of regulatory ncRNAs, were discovered in 1993 in *C. elegans* [55]. They are one of the main negative post-transcriptional regulators of gene expression. This regulation is achieved by the direct, incomplete, base pair targeting of mRNAs, which results in the recruitment of diverse RNA-decay factors, or in translation repression.

miRNAs can be generated from both intronic and exonic regions of the genome. Transcribed by RNA polymerase II as long double-stranded primary miRNAs, they are processed by Drosha RNase III enzyme and DGCR8 protein in the nucleus. Exported into the cytosol as pre-miRNA, RNase III Dicer-1 processes it into a mature ~21 nt double stranded RNAs. After strands’ separation, the so-called guide strand will associate to the Argonaute (AGO) proteins forming the RNA induced silencing complex (RISC), the main player in gene translation repression by miRNAs [56,57].

Although miRNA expression and correct biogenesis is important in the whole organism, they appear as indispensable molecules for CNS development and adult brain function [58,59]. In fact, the depletion of miRNA biogenesis-related proteins, Dicer-1 or DGCR8, strongly affects embryonic and neural development [58,60,61,62]. Chmielarz and colleagues recently reported that the specific removal of Dicer in dopaminergic neurons leads to the progressive degeneration of these cells in mice striatum and consequent locomotor affectation [63]. In the same work, the authors stimulated miRNA biogenesis using enoxacin-Dicer stimulator—in Dicer-KO cultured cells obtaining a neuroprotective effect and rescuing neuron viability [63]. miRNAs also orchestrate neural signaling processes including neurotransmitter release and dendrites and spines generation. During synaptogenesis, local mRNA translation and protein synthesis are mediated through miRNA regulation (reviewed by Hu and Li 2018 [64]). Specifically, diverse evidence points to miR-137 as an important regulator of synaptic function and pre-synaptic protein expression [65,66,67]. Together with miR-146a-5p, miR125b and miR-223, miR-137 also controls post-synaptic responses as they target different glutamate receptors [68,69]. Deep sequencing analysis revealed that miR-135 and miR-191 are involved in spine remodeling and plasticity [70]. Among others, miR-16-5p, miR-134 and miR-132 are pivotal for dendritic growth [71,72,73,74].

miRNAs have likewise been studied in glial cells. Several works on murine cells and animal models have reported the role of specific miRNAs in gliogenesis [75]. The specific deletion of Dicer1 in mouse oligodendrocyte precursors revealed the role of miR-219 in their differentiation [76]. Additionally, other miRNAs such as miR-125b and miR-124 are implicated in cerebral immune response through the regulation of glial cell proliferation and microglial polarization [77,78], as reviewed by Guo et al. [79].

### 3.2. tRNA-Derived Fragments

tRNAs-derived fragments (tRFs) are being revealed as a new class of regulatory ncRNAs. tRFs (<40 nt) are bioactive molecules classified according to their size and the part of the mature tRNA from where they derive, as previously reviewed in Kumar et al. [80]. tRFs can be constitutively produced and/or induced as a result of cellular stress response pathways. Diverse nucleases are responsible for the biogenesis of tRFs in vertebrates including RNase Z, Dicer and angiogenin (ANG). Under stress conditions, ANG produces a subtype of tRNA halves (tRNA-derived stress-induced RNAs or tiRNAs) [81], and Dicer is involved in the biogenesis of tRFS derived from the D loop or the 3′-end of mature tRNAs [82]. However, the role of Dicer in tRF biogenesis may be limited, as its depletion has little impact on tRF levels in multiple model organisms [83].

tRFs are highly abundant in the brain, where they accumulate during ageing [84,85], pointing these species as risk factors in neurodegenerative conditions. Furthermore, tRF expression dynamics have been associated to stem cell differentiation [86], denoting that the biogenesis of specific tRFs is an indicator of cell status and participates in biological processes associated to particular physiological or pathological conditions.

Although tRFs’ heterogenous functions are not entirely understood, they regulate mRNA processing and translation, controlling processes such as cell growth and differentiation [87]. The discovery of stable tRFs–AGO complexes led to the hypothesis that tRFs could work as miRNA inhibiting gene expression. Recently, a tRF derived from a tRNA–Gly with miRNA structural and functional features, was shown to bind to AGO and repress the expression of the Replication Protein A1 (RPA1) gene [88]. However, a recent large-scale meta-analysis of CLASH sequencing data (cross-linking ligation and sequencing of hybrids) [89] suggests a more complex scenario, since tRF-RNA pairing in AGO1 occurs in the 3′-UTR, 5′-UTR and intronic regions of genes. tRFs can also regulate gene expression by competitively binding to AGO proteins, thus affecting the silencing activity on target genes [90].

Furthermore, tRFs repress translation through binding to ribosomal factors, emphasizing their implication in multiple layers of gene expression regulation [91]. Specific tiRNAs, derived from tRNA-Ala and tRNA-Cys, have been shown to inhibit protein translation by assembling into a G-quadruplex-like structure (G4-motif). G4–tRFs displace eIF4G/A from caped and uncapped mRNAs, inhibit translation initiation, and induce the assembly of stress granules, which are dense cytosolic aggregates where RNA-binding proteins control the utilization of mRNA during stress [92]. However, G4-tRFs do not impair translation from internal ribosomic entry sites (IRES), underlying survival and apoptosis pathways [93,94]. tiRNAs cooperate with the translator inhibitor YB-1 to promote stress granules’ formation [95], promoting survival under acute stress conditions in U2OS or MCF7 cells. These studies highlight a mechanism by which the OS-induced tRNA cleavage inhibits protein synthesis and activates a cytoprotective stress response program. Nevertheless, while these studies have been performed in cell lines undergoing acute stress, the role of tiRNAs and stress granules in neuronal pathology involving chronic stress needs to be clarified.

### 3.3. Long Non-Coding RNAs

The ENCODE project established the existence of 9000–50,000 lncRNAs in the human genome. Transcribed from intergenic regions, from gene regulatory regions and from specific chromosomal regions by RNA polmymerase II, they are usually subjected to capping and alternative splicing [7,96]. lncRNAs’ main functions encompass the control of gene expression at different levels—both transcriptional and translational—the control of chromatin folding and the recruitment of chromatin modifiers [97,98]. Due to this wide repertoire of functions, lncRNAs can be located in the nucleus maintaining the nuclear architecture, in the cytoplasm regulating gene expression and in the mitochondria where they participate in signaling and regulate cell-energy and apoptosis [99,100]. The function of lncRNAs is associated with their unique subcellular localization [101], but how cells sort different lncRNAs to specific subcellular compartments remains unclear. Furthermore, although several lncRNAs have been well characterized to date, further research is still required to understand their myriad of functions.

Forty percent of described lncRNAs have been reported in the CNS [102], where they participate in gene expression, imprinting and pluripotency regulation, as reported by numerous studies [103,104,105,106]. One of the first described lncRNAs was H19, a paternally imprinted gene, whose importance in embryonic development and neural viability has been established [107]. Furthermore, H19 is involved in glia activation during epilepsy [108]. Other lncRNAs, such as Evf2 and MALAT1, are highly abundant in neurons showing a dynamic expression pattern in cell differentiation. Evf2 regulates neural differentiation [109] and MALAT1 plays an essential role in synapse formation and dendritic development in hippocampal neurons [104]. NeuroLNC, a nuclear lncRNA, has been recently involved in synapsis establishment. Its interaction with the well-known RNA-binding protein TDP-43 appears to be essential for the correct release of neurotransmitters [106]. In addition, RMST1 functions as a guide-RNA for SOX2 transcription factor towards neurogenic genes [110] promoting neural differentiation although it is also expressed in astrocytes [111]. Similarly, Pnky lncRNA, a highly conserved lncRNA, is very abundant in neural stem cells, regulating alternative splicing and cell fate decisions. Ramos and colleagues described it as essential for neural cell differentiation and renewal [112]. Another highly conserved lncRNA, Cyrano, is as well enriched in the nervous system, being a very important molecule for embryonic stem cells maintenance. Kleaveland et al. determined a whole functional circuit for Cyrano in which it represses miR-7 expression and avoids CDR1-AS destruction, a circular RNA known to regulate neuronal activity [113]. On the other hand, NEAT1, a lncRNA mostly expressed in oligodendrocytes and astrocytes, regulates alternative splicing and has been related to the neural response to stress [99].

### 3.4. Circular RNAs (circRNAs)

Circular RNAs (circRNAs)**,** are the only type of ncRNA with covalently linked ends. Usually derived from protein-coding genes and single-stranded, they are produced by a non-canonical form of splicing, the so-called back-splicing [114,115]. This extraordinarily stable class of ncRNAs have been found in diverse species, from prokaryotes to eukaryotes, being expressed in a tissue-specific manner. circRNAs are generally found in the cytoplasm where they mostly act as miRNA sponges [116]. Other attributed circRNA functions comprise the regulation of riboproteins and RNA interaction, the transport of miRNAs inside the cells or the regulation of cell differentiation and proliferation [116].

Similar to other ncRNAs, circRNAs are over-represented in the nervous system, showing a differential expression pattern along neurodevelopmental stages and in the distinct brain regions. As recently reviewed somewhere else [117], the high expression of circRNAs during neurogenesis and nervous system development has been already reported [117,118] and their role in synaptic transmission has been widely explored. In particular, CDR1-AS, a miR-7 sponge, was analyzed in a mouse model where synaptic malfunction and an adverse sensorimotor effect was observed after *Cdr1as* KO [119]. Moreover, hippocampal neurons present an increased expression of circHomer1_a among other circRNAs, in their post-synaptic terminals, contributing to synapses homeostasis and function [118]. Another example of the important regulatory role of circRNAs is CircDYM, shown to regulate cerebral immune response by decreasing microglial activation [120].

## 4. NcRNAs and Oxidative Stress Management in Neurodegenerative Diseases

Diverse classes of ncRNAs are involved in OS signaling pathways. The general oxidation of RNAs, including the most abundant classes of ncRNAs (rRNAs and tRNAs) is an early event in major neurodegenerative diseases (reviewed in [121]). However, the understanding of the impact of oxidized rRNA on cell physiology is still very limited [122]. Other classes of housekeeping ncRNAs (snRNAs and snoRNAs) are also altered in response to ROS accumulation [123], with specific species modulating the cell response to OS in cancer [124]. Interestingly, defects in snoRNAs are associated with neurodevelopmental disorders such as Prader–Willi syndrome [125]. Moreover, several works have identified small RNA molecules generated through the processing of snoRNAs (sno-derived RNAs), specifically involved in gene silencing [126]. Nevertheless, although snRNAs and snoRNAs are altered in neurodegenerative conditions, their direct involvement in OS management and neuronal dysfunction needs further research efforts.

Compelling evidence exists involving the regulatory ncRNAs in OS-neurodegeneration pathways. Regulatory ncRNA expression is sensitive to ROS accumulation, with these perturbations contributing to neuronal dysfunction and microglial activation. miRNAs are the most intensively studied class, and different types are up- and downregulated in response to OS; however, analogous studies regarding other classes of ncRNAs are scarce. The dynamic changes in the concentration of specific ncRNA that target important genes for the control of ROS homeostasis and metabolism, together with the intrinsic differential susceptibility of cells to ROS-damage, underlie the cell capability to cope with sustained ROS production. This is particularly relevant in neuronal cells with increased susceptibility to OS as explained in the previous section.

Specific ncRNA pathways regulating the cell response to OS are similarly altered in diverse neurodegenerative conditions, which may underlie commonly perturbed pathways, including mitochondrial dysfunction and the spread of insoluble protein inclusions. However, the altered activity of ncRNAs specifically targeting disease-related genes provides a scenario highlighting the differences in the development and progression of the different neurodegenerative diseases (Figure 2, Appendix A).

In this section, we will focus on regulatory ncRNA deregulation with direct implications in OS in the context of major neurodegenerative conditions: Alzheimer’s disease (AD), Parkinson’s disease (PD), Huntington’s disease (HD) and amyotrophic lateral sclerosis (ALS). Most studies report miRNA alterations and we will provide the latest evidence for the involvement of lncRNAs, and other less known classes.

### 4.1. Alzheimer’s Disease

AD is a progressive neurodegenerative disease associated with memory deficits and cognitive decline. AD is the most common form of dementia, accounting for 60 to 80% of cases, with less than half expected to be pure AD and the majority expected to be mixed dementias [127]. The main hallmarks of AD are the extracellular accumulation of amyloid-β (Aβ) peptides and the hyperphosphorylation of the microtubule-associated Tau protein, leading to the accumulation of senile plagues and neurofibrillary tangles, respectively. These changes are accompanied by progressive synaptic loss, and severe neuronal demise. OS plays a major role in AD, believed to be stronger than in other neurodegenerative diseases [128]. Despite the vast amount of knowledge about the pathology, the causes of AD development are still not clear. Mutations in PSEN1, PSEN2 and APP genes are associated to rare autosomal, dominant, early-onset AD; [129] and variants of the APOE gene are a strong genetic risk factor for both early-onset and late-onset AD [130].

Diverse classes of ncRNAs are involved in AD pathogenesis [131]. Numerous studies report the participation of miRNAs in AD, by regulating the target genes involved in the accumulation of Aβ and Tau phosphorylation [132,133,134,135]. In addition to these mechanisms, miRNAs regulate the cell response to OS, which is key in AD progression, causing early, chronic inflammation and underlying mitochondrial dysfunction, abnormal changes in lipids and proteins, the oxidative damage of nucleic acids and changes in gene expression [128]. Loops may be generated in which ncRNAs change their expression under OS and numerous proteins involved in OS management are targeted by miRNAs.

As previously mentioned, aging is a major risk factor in neurodegenerative conditions, where ROS are progressively accumulated [26]. In line with this, ageing-associated decreased levels of miR-186 may contribute to AD pathogenesis since miR-186 targets 3′UTR of BACE1 encoding the β-secretase enzyme that processes APP to Aβ peptides and protects from OS-pathogenesis in AD [136].

Through studying ROS-modulated miRNAs in oxidative stressed primary hippocampal neurons, a group of miRNAs (miR-329, miR-193b, miR-20a, miR-296, and miR-130b) was upregulated [137]. Analogous perturbations were found in the hippocampus of senescence-accelerated mouse strains, and enrichment pathway analysis of genes targeted by co-regulated miRNAs highlighted pathways analogously affected in AD, including apoptosis and MAPK signaling [138]. These data suggest an involvement of these miRNAs in the frame ncRNAs–OS–AD [137]. Specific miRNAs can also promote OS. For instance, miR-125b has been tagged as an important factor in AD promoting APP, BACE1 and Tau overexpression and hyperphosphorylation [139]. In a neuroblastoma cell model, the overexpression of miR-125b induced OS and stimulated apoptosis. Furthermore, supporting a link between miRNA deregulation, OS and neuronal dysfunction, miR-125b stimulated the production of pro-inflammatory cytokines [140].

In AD, ROS-generating soluble Aβ [141] induces the expression of miR-134, miR-145 and miR-210 and reduces the expression of miR-107 [142]. Decreased expression of miR-107, which targets BACE1 mRNA, is associated with early stages of AD [142]. Furthermore, carriers of APOE4 allele, show decreased levels of miR-107 [143,144] and a parallel increased production of Aβ peptides, which in turn produce Tp53 gene expression deregulation involved in cell death in neurodegenerative conditions, including AD [143]. Overall, these data suggest a positive feed-back loop in which the early decrease in miR-107 contributes to sustained ROS biogenesis in AD. In addition, miR-153 that targets APP and APLP2, an APP homologue, is decreased in a human neuronal cell line upon exposure to Aβ peptides and H_2_O_2_ [145], further linking ncRNAs with OS-pathology in AD.

ROS equally influences Tau phosphorylation and the associated neurofibrillary tangles accumulation in AD. Numerous studies correlate miRNA activity with Tau phosphorylation. For instance, miR-200a-3p reduces Aβ accumulation and Tau hyperphosphorylation by targeting BACE1 and PRKACB, respectively [146]. miR-219 is overexpressed in AD brains and in a neuroblastoma cell line, where decreased phosphorylated Tau is observed due to the miR-219 targeting of TTBK1 and GSK-3β [147]. Other miRNAs show an AD deregulation pattern that may contribute to Tau-phosphorylation pathology. miR-132-212 is amongst the top downregulated miRNAs in AD, and has been revealed as a master regulator of neuronal health, through a direct regulation of the Tau modifiers acetyltransferase EP300, kinase GSK3β, RNA-binding protein Rbfox1, and proteases Calpain 2 and Caspases 3/7 [148]. In addition, miR-132-212 directly targets Tau mRNA [133] overall suggesting that the deficiency of this miRNA contributes to neuronal dysfunction in AD and other tauopathies. Another miRNA involved in Tau metabolism is miR-26b which is upregulated in AD brains at early stages of the disease [149]. Induced miR-26b overexpression in rat brains resulted in increased Tau-phosphorylation and apoptotic neuronal death. Moreover, miR-26 blockage in cell culture was neuroprotective in front of OS [149]. Furthermore, miR-146a upregulation in AD may have detrimental consequences by targeting the ROCK1 kinase. ROCK1 decreased activity is associated with the inhibition of phosphorylation of the phosphatase and tensin homolog (PTEN), a pathway involved in neuronal Tau hyperphosphorylation in AD [150].

tRFs are another type of regulatory small ncRNAs included in the present review, with emerging roles in the physiology and pathology of the nervous system. There is little information directly correlating the activity of specific tRFs in AD pathology. However, 13 tRFs are deregulated in the senescence-accelerated mouse prone 8 (SAMP8) as an AD model and other aging related neurodegenerative diseases [85]. The pathway enrichment analysis of the mRNAs putatively targeted by deregulated small tRFs highlighted brain functions, such as vesicle synapse formation and vesicle cycle. In addition, tRFs accumulate during neuronal OS [151], a major contributor to AD pathology. These results, the fact that tRFs increase in aging in the nervous system [84,85] and the activity of specific species in the cell response to stress strongly suggests a role in AD pathology.

In addition to sncRNAs, the most recent studies show that the manipulation of diverse lncRNAs regulates the cell response to OS in cell and mouse models of AD, suggesting they should be considered in therapeutic developments. Mechanistically, lncRNAs modulate the expression of specific genes involved in cell survival and/or in the management of OS. Silencing SOX21-AS1 lncRNA has been proposed to alleviate neuronal OS and suppress neuronal apoptosis in AD mice through the upregulation of FZD3/5 and the subsequent activation of the Wnt signaling pathway [152]. Other studies have shown that silencing BDNF-AS [153] and ATB [154] lncRNAs in PC12 neuronal cell model attenuates apoptosis and OS induced by Aβ.

A common mechanism of action of lncRNAs is the specific sponging of miRNAs that results in their reduced activity. It has been shown that Aβ treatment increased the expression of XIST in hippocampal neurons. XIST knockdown ameliorated the toxicity, oxidative stress, and apoptosis induced by Aβ in hippocampal neurons through a mechanism involving miR-132 targeting [155]. In an Aβ-induced mouse model of AD, the increased expression of SNHG1 was detected, and its knockdown could reverse Aβ toxicity. Besides, SNHG1 sponged miR-361-3p, which targets ZNF217, suggesting that SNHG1 detrimental effects are the result of the deregulation of the miR-361-3p/ZNF217 axis [156]. Another recent study using an analogous AD mouse model showed that the lncRNA TUG1 that binds miR-15a was upregulated in response to Aβ intraventricular injection. TUG1 silencing and miR-15a upregulation improved the spatial learning ability and memory ability, ameliorated pathological injury, depressed neuronal apoptosis, and strengthened the antioxidant ability of hippocampal neurons in AD mice [157]. Although these studies point to a role of lncRNAs in OS-induced neurotoxicity, data on the deregulation pattern of the aforementioned lncRNAs in AD in humans are lacking and therefore the involvement of these species in AD evolution needs further investigation.

Through the regulation of miRNA availability, circRNAs provide a novel, little explored source of bioactive ncRNAs, whose dysregulation may impact neurodegenerative processes, including AD [158]. circRNA deregulation is detected in the cerebrospinal fluid of AD patients [158] and in AD cortices, showing associations with AD diagnosis and neuropathological severity [158]. Furthermore, changes were reported at pre-symptomatic stages in autosomal, dominant AD, suggesting a possible involvement in disease progression [159]. Specifically, ciRS-7 is downregulated in the grey matter of AD patients’ cortex [160,161]. ciRS-7 sponges miR-7 and its downregulation results in the increased activity of miR-7. Because miR-7 targets Ubiquitin Ligase 2A (UBE2A) involved in the autophagic clearance of Aβ [162], it is tempting to speculate about the possible involvement of ciRS-7-miR-7-UBE2A pathway in Aβ metabolism and OS. ciRS-7 has been recently shown to regulate the levels of APP and BACE1 protein by promoting their degradation via proteasome and lysosome [163]. This would point to ciRS-7 being a neuroprotective. In addition, APP reduced the level of ciRS-7, revealing a mutual regulation of ciRS-7 and APP, which would explain the reduced ciRS-7 expression in AD [163]. Another recent study shows that the expression of circHDAC9, which is a miR-138 sponge is inversely correlated with that of miR-138 in Aβ-treated N2a cells and APP/PS1 mice. Decreased miR-138 results in excessive Aβ production induced by miR-138, in vitro. The authors show decreased levels of circHDAC9 in the serum of AD patients; suggesting that the pathway circHDAC9/miR-138 could participate in APP processing in AD [164].

### 4.2. Parkinson’s Disease

PD is the most common movement disorder, affecting approximately 1% of the population over 60 years old and reaching 4–5% of the population over 85 years old. Clinically, PD is mainly described by motor symptoms, which include a tremor at rest, bradykinesia, stooped posture and a characteristic festinating walking. Non-motor symptoms such as depression, sleep disorders, and dementia precede the appearance of motor symptoms and worsen with progression of PD [165]. Pathologically, PD is defined by the loss of dopaminergic (DA) neurons in the substantia nigra (SN) pars compacta, corpus striatum and brain cortex [166]. This loss is accompanied by the presence of cytoplasmic protein inclusions, named Lewy bodies (LB), and enlarged aberrant Lewy neurites (LN). The causes leading to PD are not well understood. While genetic mutations in specific genes account for 5–10% of cases, in the majority of cases the etiology is unknown [167,168]. Familial and idiopathic forms of the disease share common pathogenic pathways, including mitochondrial dysfunction, energy production imbalance disruption of the ubiquitin–proteasome system and OS [166,169,170].

Major sources of OS in the highly vulnerable DA neurons include dopamine metabolism, the impairment of the antioxidant system, mitochondrial dysfunction and α-synuclein aggregation [170]. Excess of oxidized dopamine in the cytosol produces ROS, which was proposed to underlie selective vulnerability of DA neurons. miRNA-133b regulates cytosolic concentration of dopamine, by indirectly targeting the vesicular monoamine transporter 2 (VMAT2). miR-133b downregulates the transcription factor Pitx3, resulting in decreased VMAT2 expression [171]. The dopamine transporter DAT is another target of miR-133b that can regulate cytosolic dopamine concentration and ROS production in DA neurons [172]. Its downregulation in the PD midbrain suggests its possible participation in PD neuropathology and OS [173].

Diverse genes involved in the correct function of mitochondria are mutated in familial forms of PD, including DJ-1, Parkin and PINK1 [169]. However, altered levels or the activity of normal versions of these genes underlie mitochondrial dysfunction and OS also in sporadic, non-familial cases of PD. In this scenario, perturbations of ncRNAs that target these genes may contribute to PD neuropathology. Precisely, PINK1 protects neurons from OS [174] and its loss of function increases ROS biogenesis in dopaminergic neurons [175]. miR-27a and miR-27b target PINK1 and regulate the autophagic clearance of damaged mitochondria [176]. Furthermore, miR-27a regulates the expression of subunits of the mitochondrial complex [177], overall suggesting that their manipulation could be considered in therapeutic designs.

DJ-1 protein is an important player of mitochondrial activity and cell response to OS [178,179,180]. miR-4639 inhibits DJ-1 expression, and subsequent OS and neuronal death. Early miR-4639-5p upregulation has been recently shown in the plasma of PD patients [181], thus identifying this miRNA as a putative biomarker and a target for therapeutic development. Moreover, a recent study has shown that DJ-1 downregulation results in miR-221 decreased expression in mouse and cell models [182]. miR-221, with a relevant role in neurite outgrowth and neuronal differentiation, regulates the expression of diverse pro-apoptotic proteins and prevents OS-induced BIM expression. In addition, miR-221 protects neuronal DA cells knocked down for DJ-1 from MPP+-induced OS and cell death. Together, these data point to an important role of miR-221 perturbed expression in the neuronal response to OS. Additionally, the upregulation of miR-494 contributes to OS-induced neuronal death by inhibiting the expression of DJ-1 [183].

The downregulated expression of miR-34b and miR-34c has been likewise reported in PD brains, at pre-motor, early stages of PD [184]. The reduced activity of miR-34b and miR-34c using a complementary antisense oligonucleotide in a DA neuronal cell line resulted in ROS induction, mitochondrial dysfunction and cell death, and a parallel decreased expression of Parkin and DJ-1. The deregulation at pre-motor stages of PD brains can be extended to diverse classes of ncRNAs [185], suggesting that ncRNA perturbations occur early in the course of the disease, and participate in disease evolution.

In PD, deregulated miRNAs may also contribute to an impaired antioxidant system. OS induces the translocation of the transcription factor Nrf2 to the nucleus, where it activates the expression of genes involved in the response to OS. Keap-1 is an inhibitor of Nrf2 and is regulated by diverse miRNAs, including miR-153, miR-27a, miR-142-5p, and miR-144 and miR-7 [186,187,188,189]. Among them, miR-7 downregulation is of particular interest since it has been shown to be decreased in the substantia nigra of PD patients [187]. In PD, decreased miR-7 may contribute to the impaired expression of antioxidant genes, through the Keap-1, Nrf2 pathway.

A recent study showed that, in addition to miRNAs, diverse types of sncRNAs are deregulated in PD human brains [185]. Among the different classes, tRFs were revealed as the species optimally classifying pre-motor cases of PD versus control patients; suggesting that tRFs are early perturbed in PD. The mechanisms underlying the altered expression and activities of deregulated tRFs have not been elucidated. ANG is directly involved in the stress-induced biogenesis of tiRNAs [81]. Specific ANG mutations have been found in PD patients [190], and ANG is decreased in a α-syn mouse model of PD [191], suggesting that ANG dysfunction could contribute to tRFs perturbations. Mechanistically, ANG-generated tiRNAs have been proposed as protective [90]; however, the virally mediated overexpression of ANG in the substantia nigra failed to protect from DA neuronal loss in a neurotoxin-based mouse model of PD [192]. Other studies confirm a link between ANG-induced tiRNAs and the neuronal response to stress [193]. The lack of NSun2 that methylates two cytosine residues in tRNAs results in an increase in stress-induced ANG-mediated 5′-tiRNAs accumulation. The increase in these species contributes to the death of hippocampal and striatal neurons, through the inhibition of protein translation and cell stress [193]. These studies suggest a context-dependent effect of ANG–tiRNAs as neuroprotective or neurotoxic molecules. Moreover, a high-throughput functional analysis, that included antisense oligonucleotides targeting specific tRFs deregulated in PD brains, identified specific types involved in the response of dopaminergic SH-SY5Y cells to MPP+ OS [194]. In brains from motor cases of PD, a tRNA-Tyr derived fragment is upregulated [185]. This tRF sensitizes neuronal cells to OS-induced p53-dependent cell death [195], suggesting a possible contribution of TyR-tRF biogenesis to the neuronal dysfunction in PD.

On the other hand, numerous studies have recently highlighted the participation of lncRNAs in OS pathways in the context of PD. MPTP (1-methyl-4-phenyl-1,2,3,6-tetrahydropyridine) and paraquat, that damage the substantia nigra and are widely used to model PD [196], influence the activity of the transcription factor Nrf2 and alter the expression of diverse lncRNAs. In a rat model of PD induced by 6-OHDA, the lncRNA UCA1 was increased in the rat midbrain [197]. The siRNA-mediated decrease in UCA1 upregulated the expression of tyrosine hydroxylase (TH) positive cells, reduced apoptosis and OS in neurons of the substantia nigra and improved the neuroinflammatory response [197]. Several works report the role of MALAT1 lncRNA in apoptosis regulation in PD pathology. The downregulation of MALAT1 in MPP+-induced animal and cell models inhibited apoptosis through the sponging of miR-124 [198]. TUG1 that sponges miR-152-3p is increased in mouse and cell models of PD [199]. In these models, TUG1 silencing protected dopaminergic SH-SY5Y cells and neurons from OS and neuroinflammation. In cells submitted to MPP+ stress, TUG1 inhibition was accompanied by a suppression of PTEN and cleaved caspase-3 expression and an increase in TH. The protective effects of TUG1-decrease were reversed in the presence of a miR-152-3p inhibitor, suggesting that the miR-152-3p/PTEN pathway regulated TUG1 effects.

In addition, the linc-p21 is upregulated in PD [200], and highly expressed in SH-SY5Y cells undergoing MPP+ stress [201]. The knockdown of linc-p21 attenuated cytotoxicity, OS, cell apoptosis and neuroinflammation induced by MPP+ in this model. Moreover, the overexpression of linc-p21 produced opposite effects, overall suggesting that linc-p21 is an OS responsive and dosage sensitive lncRNA likely involved in PD. The linc-p21-regulated response to MPP+ depends on its activity as a sponge of miR-625, which in turn regulates TRPM2 expression. In the substantia nigra of PD patients and in diverse PD model systems, the neuron-specific lncRNA NEAT1 is also upregulated [202]. However, NEAT1 expression has been shown to be neuroprotective, since the interference-mediated depletion of NEAT1 aggravated death in the cells exposed to paraquat in a LRRK2-mediated manner. A study focused on identifying deregulated lncRNAs in the substantia nigra of PD patients described a pronounced increase in the Long Intergenic Non-Protein Coding RNA, p53-Induced Transcript, LINC-PINT, that was also observed in other models of PD and other neurodegenerative conditions [203]. Experimentally, the depletion of LINC-PINT exacerbated the death of cultured neuronal cells exposed to OS, suggesting a neuroprotective role of this transcript in diverse neurodegenerative disorders.

Studies about the possible involvement of circRNAs in PD are limited. In the SN, the age-dependent accumulation of circRNAs observed in healthy individuals was lost in PD individuals [204]. However, in the same brain area, increased levels of CircSLC8A1 were found, with this circRNA showing binding sites for miR-128. Hence, miR-128 targeted genes were increased in the PD SN, suggesting a regulation of miR-128 activity by CircSLC8A1. In addition, CircSLC8A1 expression was sensitive to OS status in culture cells, overall suggesting an OS-mediated deregulation of the CircSLC8A1/miR-128 pathway in PD. circRNAs have also been shown to be deregulated in multiple brain regions in the MPTP mouse model of PD [205]. The enrichment pathway analysis of deregulated circRNAs highlighted diverse biological functions, including synapse function, neuronal differentiation, axon guidance and PD metabolism. Specifically, the authors suggest that the mmu_circRNA_0003292-miRNA-132-Nr4a2 pathway can be involved in the regulation of pathogenic mechanisms [205]. Using the same mouse model of PD, a recent report has shown a decreased expression of circDLGAP4 upon MPTP treatment and an analogous deregulation pattern in MPP+ treated cell models [206]. In SH-SY5Y and MN9D cells, circDLGAP4 was neuroprotective, exerting its function via the regulation of miR-134-5p activity, which in turn regulates the cAMP response element-binding protein (CREB). The circDLGAP4/miR-134-5p axis modulates the CREB-dependent signaling which affects the expression of genes involved in neuronal survival, providing the mechanistic scenario for the protective activity of circDLGAP4 and the detrimental consequences of its decrease in PD. A direct role of ciRS-7 in PD has not been established. However, the ciRS-7 targeting of miR-7 activity results in increased levels of miR-7-regulated genes, including α-syn [161]. Another recently reported sponge of miR-7 is circSNCA [207]. In the MPP+ PD cell model, the dopamine receptor agonist pramipexole, used in PD treatment, downregulated circSNCA, resulting in an miR-7 increase, an α-syn decrease and reduced cell apoptosis. These results point to a better understanding of PD pathology and to possible therapeutic designs contemplating the manipulation of circRNA activity.

### 4.3. Huntington’s Disease

HD is a dominant rare hereditary disorder (1:10,000–1:20,000), caused by an expanded CAG trinucleotide repeat in the exon 1 of the HTT gene (reviewed in McColgan et al. [208]). CAG repeat length correlates with age onset symptoms that include motor, cognitive psychiatric alterations, combined with sleep and circadian disturbances. A major neuropathological feature is the loss of striatal GABAergic medium spiny neurons and the cortical neurons projecting to them, accompanied by astrogliosis and microglia activation. The progressive atrophy of the striatum and cerebral cortex leads to patient death at 15–20 years from the disease onset [208,209].

The expanded CAG repeat produces an abnormal protein with an expanded track of glutamines that has been proposed to underlie a number of toxic effects [209,210]. In addition, the RNA has been recently highlighted as a direct pathogenic contributor, involving gene silencing, the sequestrations of proteins with affinity for the CAG repeat RNAs and repeat associated non-ATG (RAN) translation resulting in homopolymeric proteins prone to aggregation [211,212].

Strong miRNA deregulation is detected in brain samples of HD patients and HD mouse models. In humans, miRNAs and miRNA-variants (IsomiRs) are dysregulated both in the frontal cortex and putamen, the most affected areas in HD [213]. Although the effect of the isomiRs is uncertain, the perturbations of those involving the seed region of the miRNA may likely vary the gene targeting profile, thus increasing the complexity in the miRNA depending effects. The strong perturbations in miRNA expression described in HD could be explained, at least in part, by the accumulation of AGO2 in aggregates [214], through a mechanism involving impaired autophagy [215]. AGO2 is a major executor of the miRNA-mediated gene silencing, and its accumulation in HTT-aggregates results in abnormal miRNA activity and expression levels. Furthermore, reduced levels of Drosha and Dicer, involved in the biogenesis and activity of miRNAs, are downregulated in mouse models of HD [216], providing further evidences for a general dysregulation of miRNA pathways. Specifically, Packer et al. reported that miR-9/miR-9* is downregulated in HD, with this resulting in the mislocalization of the transcriptional repressor REST that controls the expression of neural genes and is a key mediator of the transcriptional changes occurring in HD [217]. Specific miRNAs are altered at pre-motor stages of the disease in a knock-in mouse model, indicating that miRNA perturbations are an early event [218]. This study showed CAG length-dependent miRNA expression changes in the brain, especially in the striatum, revealing a molecular signature that correlates with the onset of disease. In comparing mRNA and miRNA transcriptomic data from the same animals, the authors highlighted a number of miRNAs with potential relevance in the regulation of HD mRNA perturbations, including miR-212/miR-132, miR-218 and miR-128 [218].

HD-perturbed miRNAs have been shown to regulate the cell response to OS. For instance, the sequence-specific inhibition of miR212/miR132 induces apoptosis in cultured primary neurons, whereas their overexpression is neuroprotective against OS in neurodegenerative scenarios [219]. In addition, miR-218 downregulation protects from oxidative-glucose deprivation/re-oxygenation in PC12 cells, through reducing inflammatory cytokines secretion, OS status, and apoptosis rate [220]. Furthermore, a number of the HD-deregulated miRNAs have been identified as general regulators of the neuronal response to OS, in a high-throughput functional screening [194]. These data strongly suggest a contribution of miRNA perturbations in the OS management, linked to HD.

lncRNA involvement in OS in the context of HD has been little explored; nevertheless, specific lncRNAs are perturbed in HD and likely contribute to pathogenesis. An analysis of microarray data reported diverse deregulated lncRNAs, including DGCR5, a target of the transcriptional repressor REST [221]. In addition, the upregulation of TUG1 and NEAT1 and the downregulation of MEG3 are detected in the HD caudate [221]. The upregulation of the TUG1 and NEAT1 mouse orthologs has been validated in the brain of diverse HD mouse models [222]. The inhibition of NEAT1 and MEG3 results in a significant decrease in HTT aggregates and Tp53 expression inhibition in a cell model. Although an OS readout is not provided, alterations in Tp3 and protein aggregation have direct links with cell OS.

### 4.4. Amyotrophic Lateral Sclerosis

ALS is a progressive disorder characterized by the progressive degeneration of motor neurons in the brain and spinal cord leading to the loss of the voluntary control of movements. It begins with focal weakness but spreads inexorably to involve most muscles. Death due to respiratory paralysis typically occurs in 3 to 5 years. The total number of cases is approximately 3 to 5 per 100,000 (reviewed in [223]). The degeneration of motor neurons is accompanied by neuroinflammatory processes, with the activation and proliferation of astroglia, microglia, and oligodendroglial cells [224,225]. SOD1 mutations are present in about 15 to 20% of families with ALS [226]. In addition, a hexanucleotide repeat expansion in the first intron/promoter region of the lncRNA C9ORF72 is the most common genetic cause of ALS, present in 40% of familiar cases [227]. Mechanistically, the RNA toxicity of C9ORF72-related ALS is based on the direct sequestration of important RNA binding proteins by the hexanucleotide repeat and RAN translation into dipeptide repeats.

Familial and sporadic ALS forms commonly show the aggregation of specific proteins in the cytoplasm, especially in motor neurons. These include the nuclear TAR DNA-binding protein 43 (TDP-43), showing cleaved forms, hyperphosphorylation, and mislocalization to the cytoplasm [228]. Ubiquilin 2 aggregates are also common [229], similar to intracytoplasmic deposits of wild-type superoxide dismutase 1 (SOD1) in sporadic ALS [230]. The SOD1 gene encodes the cytosolic copper- and zinc-dependent superoxide dismutase (Cu, ZnSOD) that converts O^2-^ into H_2_O_2_; however, several observations make it unlikely that motor neuron death occurs because of a loss of dismutase function [231] and it is generally accepted that mutant Cu, ZnSOD presents a novel, cytotoxic activity of unknown nature. However, diverse evidence supports the involvement of OS in ALS pathology (reviewed in [231]).

Studies of miRNAs in ALS have been centered in biomarker discovery [232,233,234,235,236,237] and miRNA activity manipulation for therapeutic designs [234,236,238,239]. For example, miR-27a, miR-34a, miR-155, miR-142-5p, and miR-338-3p have been proposed as biomarkers and potential ALS therapeutic targets, directly or indirectly involved in OS [234,236,238,239]. A study identified 15 deregulated miRNAs in ALS and induced pluripotent stem cells differentiated to motor neurons [235]. In this study, gene ontology and molecular pathway enrichment analysis indicated that the predicted target genes of the deregulated miRNAs are involved in neurodegeneration-related pathways. miR-34a and miR-504 are especially interesting as they are involved in the p53 pathway. Specifically, miR-34a targets SIRT1, previously reported as involved in neural survival in an ALS mouse model [240]. In addition, a microarray analysis identified the upregulation of miR-29a, miR-29b and miR-338-3p in ALS brains [241]. miR-338-3p deregulation is particularly interesting since it targets nuclear-encoded mitochondrial mRNAs encoding subunits of the oxidative phosphorylation machinery [242]. A general downregulation of miRNA levels is found in multiple forms of human ALS. ALS-causing mutations are sufficient to inhibit miRNA biogenesis. The underlying mechanism involves stress granule formation and perturbations in Dicer and AGO2 interactions with their partners. Enhancing Dicer activity with enoxacin appeared to be beneficial for neuromuscular function in ALS mouse models, suggesting that targeting miRNA activity in future therapeutic designs would be interesting [243].

ER stress leading to apoptosis has been strongly associated with motor neuron degeneration in ALS. Nrf2, involved in redox reactions, participates in ER stress-induced apoptosis and together with ATF4 repress miR-106b-25 cluster. It has been shown that the Nrf2/ATF4/miR-106b-25 cluster may be relevant for ALS pathogenesis [244]. In addition, miR-142-5p, that is downregulated in the CSF of ALS patients, has been shown to reduce OS via the upregulation of the Nrf2 signaling pathway [236,245].

The involvement of tRFs in ALS is supported by ANG mutations occurring in familial and sporadic forms of ALS [246]. ANG mutations are associated to a decreased ANG ribonuclease activity and impaired formation of protective 5′-tiRNAs that inhibit the translation initiation and trigger the assembly of stress granules, as explained in Section 3.2 [247,248]. In a YB-1-dependent manner, cells treated with 5′-tiDNA-Ala modestly, but significantly, enhanced motor neuron survival in response to OS [93]. These data suggest that ANG dysfunction and the impaired formation of specific tiRNAs are involved in motor neuron dysfunction. However, the effect of other tRFs formed in the ALS context needs to be evaluated in more detail. A recent study has defined a serum-based ncRNA diagnostic biomarker signature containing two tRFs [249]. Whether these changes are reflected in motor neurons and the functional consequences of its perturbed biogenesis needs further research.

The mutation of the lncRNA C9ORF72 in familial cases of ALS underlie OS and senescence in induced pluripotent stem cells differentiated to astrocytes [250]. Mutant astrocytes showed the downregulated secretion of several antioxidant proteins and, conditioned media from C9ORF72-astrocytes increased OS in wild-type motor neurons. Other lncRNAs including NEAT1 and NEAT1_2 have also been related to ALS. The binding of NEAT1_2 to TDP-43 and FUS/TL, which are ALS-associated RNA-binding proteins, is an early phenomenon in ALS pathogenesis [251]. Electron microscopy showed a specific assembly of NEAT1_2 lncRNA around the interchromatin granule-associated zone in the nucleus of ALS spinal motor neurons. These results suggested that the lncRNA NEAT1_2 may act as a scaffold of RNA binding proteins in the nuclei of ALS motor neurons, thereby modulating their functions during the early phase of ALS. NEAT1, has been recently shown to regulate TDP-43 function in the response to arsenite-induced stress [252]. The formation of TDP-43-nuclear bodies in response to stress alleviates TDP-43-mediated cytotoxicity. TDP-43 nuclear bodies partially colocalized with paraspeckles, in which NEAT1, which is upregulated in stressed neurons, acts as a scaffold. NEAT1 promotes TDP-43 separation in vitro. The authors provide evidence that ALS-associated TDP-43 mutation impairs the NEAT1-mediated TDP-43 separation, with this resulting in the translocation of TDP-43 to the cytoplasm to form stress granules. Overall, these data suggest a stress-protective role for TDP-43 nuclear bodies, regulated by NEAT1.

## 5. Conclusions

The above reviewed results emphasize the relevance of regulatory ncRNAs as bioactive molecules regulating the management of OS in neurodegenerative conditions. Nevertheless, we are aware that additional ncRNA classes present in the CNS can be affected by and associated to OS and neurodegeneration, including highly abundant rRNAs. 

The overall conclusions from the works presented here are: (1) OS plays a major and early role in neurodegenerative diseases; (2) regulatory ncRNAs expression and activity dynamics regulate, directly and/or indirectly, gene expression networks involved in ROS homeostasis and buffering; (3) miRNAs are the best known class of ncRNAs, but other types of regulatory ncRNAs, including lncRNAs, tRFs, and circRNAs, appear as novel classes with a direct impact in OS modulation and neurodegeneration.

These evidences have boosted pre-clinical research, encouraging the modification of the activity of ncRNA in therapeutic designs to control OS [253,254]. This involves the use of anti-sense oligonucleotides targeting ncRNAs for their inhibition, or ncRNA mimic molecules for their activation. However, there is a number of experimental and technical considerations that need to be fully controlled for translational development, including the use of appropriate control sequences and the effective concentration of an oligonucleotide to be used for in vivo experiments. In addition, results obtained with mimic molecules should be interpreted with caution, since they are used at diverse orders of magnitude higher than normally observed in vivo, and can produce unspecific effects [255]. Experimental research to understand the functions of specific ncRNAs in OS, particularly miRNAs, has been done in neuronal or glial cell lines and non-human animal models of neurodegenerative diseases. However, cell-type-specific ncRNA profiles in immortalized, primary cultured cells or animal models differ [256], and analogous differences may be found for the ncRNA-targeted genes.

In addition, caution should be taken in translating the results from cell and animal models to the human context. Many neuroprotective treatments successfully working in rodent and primate models, have failed in human patients [257]. Furthermore, due to the heterogeneity and complexity of neurodegenerative disorders, animal models reproduce particular features of the disease process but often fail to fully recapitulate neuropathology, including the characteristic spatial pattern of neuronal loss [258]. While the manipulation of specific ncRNAs in these models produces a particular OS readout, analogous results in the human cells of the nervous system should be validated. Evolutionary conserved ncRNAs may analogously regulate OS pathways in human cells. However, the human nervous system contains the largest diversity of ncRNAs, with many human-specific species showing a highly precise expression pattern in particular neuronal and glial cell types [259,260,261]. Moreover, the functional interaction between different classes of ncRNAs (for instance lncRNAs or circRNAs buffering specific miRNAs) provides an additional layer of complexity, in which multiple ncRNA-signaling pathways orchestrate the response to OS.

Patient-derived induced pluripotent stem cells (iPSCs) differentiated to specific neuronal populations have been implemented as a human cell model to study biochemical and molecular aspects of neurodegeneration [262,263]. However, differentiation protocols are challenging [262] and cultured neurons in artificial media may present differences in ncRNA expression and the regulation of OS. Three-dimensional human brain organoids offer a hopeful strategy to study OS and neurodegeneration since they recapitulate the interaction between different cell types, including glial cells known to modulate the effects of OS [264]. The development of experimental strategies to reliably monitor levels of oxidative damage in human iPSCs-derived cell models or brain organoids [265] should help in understanding the temporal dynamics of ncRNA activity modulation, OS, mitochondrial dysfunction, protein aggregates and cell survival.

Studies aimed at understanding the involvement of ncRNAs in neurodegeneration have focused on (1) identifying disease-deregulated ncRNAs and (2) reproducing the deregulation pattern in cell and/or animal models to evaluate possible detrimental consequences. The identification of ncRNA-deregulated species at pre-symptomatic stages is particularly interesting, as their perturbed activity can be an early pathogenic driver that may be targeted in therapeutic designs. Nevertheless, the majority of these studies are focused on a particular ncRNA, and do not take into account that multiple species are functionally altered in neurodegenerative diseases. This raises the challenge of targeting the activity of multiple ncRNAs to better protect from OS damage.

Another layer of activity of ncRNAs is produced by their release to the extracellular space both within vesicles, free or bound to RNA binding proteins, rendering these molecules as paracrine signaling mediators [266]. Although the majority of studies on the activity of extracellular ncRNAs have been centered in miRNAs, the most recent reports point to additional species as bioactive extracellular molecules, including lncRNAs and tRFs. In neurodegenerative diseases, the profiling of extracellular ncRNAs in biofluids has been pointed as a new source of biomarkers [267,268]. However, a new field of study is required to understand the modulation of ncRNA release and their regulatory activities in different aspects of neuropathology, including OS management.

## Figures and Tables

**Figure 1 antioxidants-09-01095-f001:**
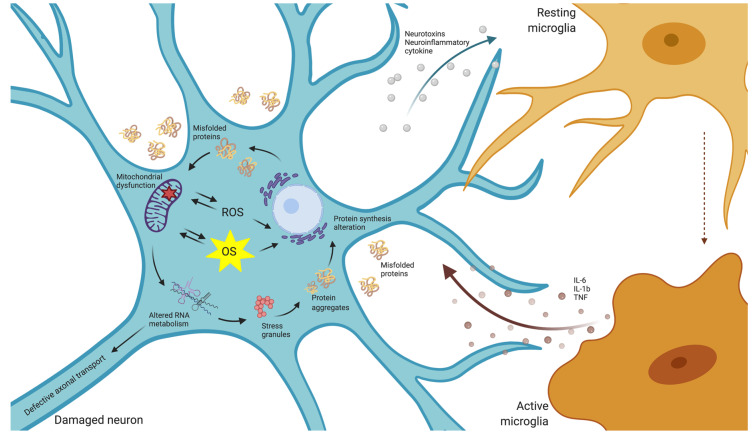
Oxidative stress in neurodegenerative disorders. Dysfunction in neural protein synthesis and processing entails mitochondrial defects, the alteration of RNA metabolism and the deposition of protein aggregates, which ultimately, encompass the generation of ROS and OS. At the same time, ROS production implies the alteration of protein folding and the production of neurotoxins which activates neuroimmune cells and enhances the production of neuroinflammatory cytokines.

**Figure 2 antioxidants-09-01095-f002:**
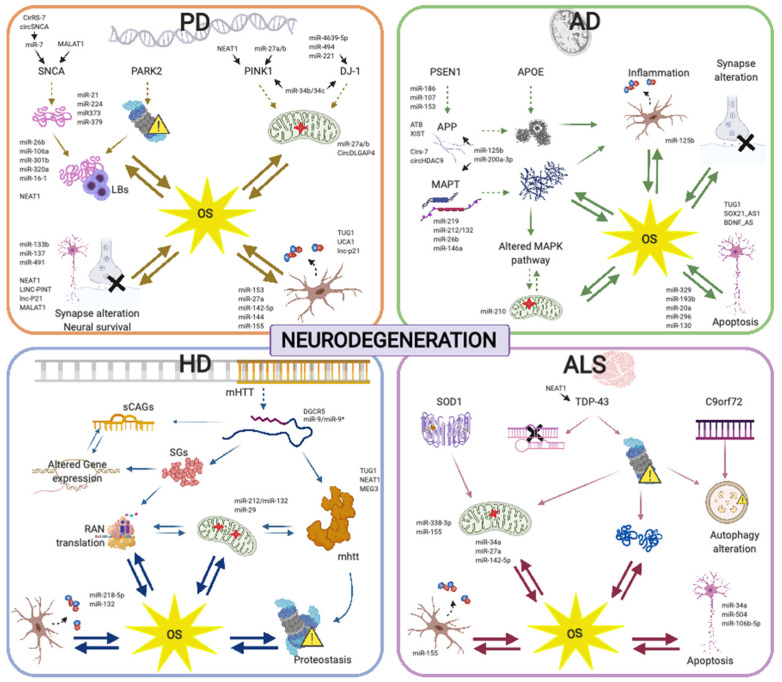
Regulatory ncRNAs are involved in oxidative stress (OS) management in diverse neurodegenerative diseases. Experimentally evidenced ncRNAs related to proteosomal and mitochondrial dysfunction, protein metabolism, reactive oxygen species (ROS) and OS generation and neural viability for Alzheimer’s disease (AD), Parkinson’s disease (PD), Huntington’s disease (HD) and amyotrophic lateral sclerosis (ALS) are shown.

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
