# Peer review of "Non-Coding RNAs as Sensors of Oxidative Stress in Neurodegenerative Diseases"

_antioxidants, 2020, doi:10.3390/antiox9111095_

Round 1

Reviewer 1 Report

Oxidative stress is one of the major contributing factors in many neurodegenerative diseases such as Alzheimer’s disease and Parkinson’s disease among others. Therefore, approaches to reduce oxidative stress have therapeutic value. This review article provides a detailed literature survey of oxidative stress in neurodegenerative diseases. The authors specifically emphasis on role of regulatory noncoding RNA in oxidative stress pathway. This review article is relevant to the field of neurodegenerative diseases.

Author Response

We thank the reviewer for the general, positive comment on our manuscript. We agree in that it may be of interest for researchers working on the molecular basis of neurodegenerative conditions, since it outlines the main concepts and ideas linking oxidative-stress- neurodegeneration and regulatory non-coding RNAs.

Reviewer 2 Report

In this review the authors summarize aspects of oxidative ncRNA damage and the possible connection to neurodegenerative diseases. This topic is timely, interesting and new developments have been reported in the recent years. This review article however falls short of making a convincing case. Major changes are needed to convert this manuscript into an acceptable review form.

1) In general this review is lengthy and hard to read. The main reasons for this are: (i) the very repetitive nature of statements and sentences. There are so many examples for this that listing them all individually would be sprawling. I strongly recommend the authors to critically read the text and mark and eliminate repetitive text (e.g. lines 12-15 compared to lines 35-37; lines 88-89 compared to 93-95; lines; etc, etc); (ii) the text also needs English proofreading. In its current form the reviews is too long and hard to digest for the reader.

2) It remains unclear why the authors solely focus on four ncRNA classes in their article. It is clear that the authors have an obvious affinity for miRNA biology, but this class of ncRNA is by far not as central for cell metabolism and oxidative damage as this review seems to suggest. Thus the selection of ncRNA classes appears almost arbitrary. The authors need to explain why they, for example, ignored the most abundant and arguably the most central ncRNA class in their review manuscript, namely the rRNAs. Multiple reports and even functional data exist that link oxidized rRNA/ribosomes to neurological diseases. Furthermore, it is also unclear why snoRNAs have been omitted. Some snoRNAs have even been linked to neurological disorder (doi: 10.1002/wrna.1417). On the other hand the authors summarize the few publications about circular RNA in quite some details, even though they do play rather a supporting role in RNA biology (at least from what we know so far). Thus the review article is not well balanced.

3) lines 229-231:  These sentences emphasize the miRNA-focus of the authors. It is simply not entirely correct that tRNA-fragments shorter than 30 nucleotides act like miRNAs. See for example a recent special issue (RNA Biol., Vol. 17, Issue 8, 2020) about latest developments in the field of tRF research.

4) the authors need to scan the entire text for consistency issues. Sometimes its “non-coding RNA” and sometimes “noncoding”; or: “Nrf2” and “NRf2; or: “linc-p21” and “linc-P21”;

5) lines 114-115: please clarify! Almost all cells “largely depend on oxidative phosphorylation”.

Author Response

We thank the reviewer the time and effort invested in the review of our manuscript. We want to note that the manuscript has been re-reviewed to address the comments of all reviewers and the text in this novel version has been substantially modified. We provide a point-to-point response to the comments of the reviewer, and performed modifications accordingly, which are next specified.

  1. We have eliminated repetitive statements throughout the text, summarizing the messages in reorganized non-recurrent paragraphs. We have also carefully revised English of the text, as requested.
  2. The reviewer wonders why we have not included information on other important classes of ncRNAs such as rRNAs tRNAs and snoRNAs. In this review we have focused in the regulatory class of ncRNAs that cover the axis OS-ncRNAs-neurodegeneration. Our aim was to cover the available information on disease-deregulated ncRNAs that manage the OS response in neurodegenerative conditions; and this is essentially the case for regulatory ncRNAs (miRNAs, lncRNAs and circRNAs). In the new version of section 3, we have provided balanced information on the chosen classes of regulatory ncRNAs. Differing from other reviews, we have also included tRFs as an abundant class of regulatory sRNAs that recent reports show to be strongly deregulated in neurodegenerative conditions and with important roles in the management of OS. These justifications have been included in the new version of the manuscript (section 3, page 4, lines 152-163).                         For space reasons we cannot cover extensive information on the effects of housekeeping ncRNAs (rRNA, tRNA, snoRNA and snRNA), which would require a separated review manuscript. Although we focus on regulatory ncRNAs, we agree with the reviewer that the highly abundant housekeeping rRNA, tRNA and snoRNAs that are responsive to OS and OS management are worth to mention. We have therefore added a paragraph at the beginning of section 4 lines 296-306.                                                                        Finally, in section 4 that addresses the axis regulatory ncRNA-OS-neurodegeneration; it is difficult to shorten the information on miRNAs for which the vast majority of studies exist. However, in the new version we have reorganized in a more comprehensive and balanced way, the information on miRNAs and other ncRNAs.
  1. The fact that tRFs can functionally work as miRNAs was an initial hypothesis born from the observation that AGO proteins were abundantly loaded with miRNAs and tRFs. In fact, loading of tRFs to AGO and the sequence-guided specific silencing has been shown. However, the reviewer is right in that this mechanism cannot be generalized, and the latest studies suggest a more complex regulation by AGO-tRFS complexes. We have included these observations (and the reference commented by the reviewer) in the new version (pages 5-6, lines 221-229)
  2. Consistency issues, including “non-coding RNA” vs “noncoding RNAs”, “Nrf2” vs “NRf2; and “linc-p21” and “linc-P21” have been corrected throughout the text.
  3. The sentence has been reformulated to “Compared to other cells, neurons are more dependent on mitochondrial oxidative phosphorylation, to fulfill their high-energy demands” (Page 3, lanes 108-109).

Reviewer 3 Report

The manuscript addresses the interesting relationship between non-coding RNAs and oxidative stress in neurodegenerative diseases. This is a well-structured review that describes the growing evidence on the role of oxidative stress in deregulation of ncRNAs in major neurodegenerative diseases. In addition to its role in the pathophysiology of these diseases, it is especially relevant at pre-symptomatic stages since i) their perturbed activity may be targeted in therapeutic approaches and, ii) can be used as a biomarkers.

Few questions should be addressed:

  1. In general, paragraphs are too short. Every few sentences, a new paragraph starts. This makes the reading less fluent and, in some cases, topics are not well connected. This occurs throughout the entire manuscript, but is especially evident in sections 1. Alzheimer’s disease and 4.2 Parkinson’s disease.
  2. Stress granules should be defined. They appear in Fig. 1 and several times through the text. An explanation for non-expert readers is advisable.
  3. Figure 1 indicates: “Defected axonal transport”. This should be corrected to “Defective axonal transport”.

Author Response

We thank the reviewer for the positive comment on our manuscript.

We have addressed the comments raised, and modified the manuscript accordingly.

  1. We have shortened the paragraphs throughout the manuscript, with special emphasis in Alzheimer’s and Parkinson’s disease sections. In addition some paragraphs have been reorganized to better connect topics. We agree with the reviewer that with these changes, reading is more fluent
  2. Stress granules have been defined , when first mentioned in the text (page 6, line 234).
  3. We have corrected “Defected axonal transport” to “Defective axonal transport” in figure 1 caption

Round 2

Reviewer 2 Report

The revised version of the review manuscript submitted by Gámez-Valero has improved. Most, but unfortunately not all, points raised during the first round of peer review were adequately addressed. For my taste the review is still lengthy and lacks a certain “flow”. This is, when reading the other two referee reports, obviously solely my personal concern. Points 2 and 3 of my previous report have been reasonably addressed and the text is now more logical and sound. The issues about typos and inconsistent abbreviations, however, is still not adequately addressed. The author’s statement in their rebuttal letter “We have also carefully revised English of the text, as requested” is already falsified in the abstract: line16: “pot-transcriptional gene expression…” I really do not regard my role as referee in spotting all typos, grammar mistakes and inconsistencies in the text. There are still numerous (too many to list them all here) cases of textual errors, inconsistencies and typos:

e.g. lane 44: “NcRNAs have been….”; lane 47: “ncRNAs are….”

e.g. lane 170: “miRNAs (…), the most….”; lane187: ”MiRNAs also orchestrate…”

e.g. lane 440: “60 years old”;   “85-years old”

e.g. lane 610: “upregulation”; lane 611: “up regulation”; lane 649: “downregulation”

grammar: lane 51: “focused on microRNAs” not “focused in micoRNAs”

typo: lane 142: “descried” instead of "described"

grammar: lane 167: “we have focused on main regulatory ncRNAs…” not “we have focused in main regulatory ncRNAs…”

incorrect abbreviation: lane 175: its RNase not RNAse

typo: line 229: “capped” not “caped”

unclear sentence: line 434-435

etc, etc, etc

Author Response

Thank you again for your comments. With this revision we believe that the manuscript has substantially improved.

In order to detect all typographical and grammatical misspelling, the manuscript has been evaluated by other colleagues who have detected errors we did not notice during the first review. 

Please find that, specifically:

  • The whole manuscript has been written and revised using American English and, words in UK English have been changed, e.g. "capped" "signaling"
  • ncRNAs, miRNAs, tRFs, circRNAs, etc are always beginning with lower case letter in this new version of the manuscript. Similar changes have been performed in Suppl Table 1 (new version also submitted)
  • "Up/down-regulat(ed/ion)" is now spelled with "-"
  • "Overexpression/overexpressed" has been used as a single word
  • All "focused in" misspelling have been corrected to "focused on"
  • typographical error missing 1-2 letters have been also solved
  • Abbreviations, such as AGO or Dicer, have been consistently changed
  • RNase has also been corrected
  • Some complex sentences have been simplified or divided in two straightforward ones.
  • Grammar error involving verbs and pronouns have also been revised, e.g."have/has"
  • As well, previously referred and defined abbreviations such as "dopaminergic neurons, substantia nigra, or central nervous system" have been further used along the manuscript as "DA, SN, or CNS"